# Valley: Video Assistant with Large Language Model Enhanced Ability

## Abstract

Large Language Models (LLMs), with remarkable conversational capability, have emerged as AI assistants that can handle both visual and textual modalities. However, their effectiveness in joint video and language understanding has not been extensively explored. In the paper, we introduce *Valley*, a multi-modal foundation model that is designed to enable enhanced video comprehension and instruction-following capabilities. To this end, we construct two datasets, namely '*Valley-702k*' and '*Valley-instruct-73k*', to cover a diverse range of video-text alignment and video-based instruction tasks, such as multi-shot captions, long video descriptions, action recognition, causal inference, etc. Then, we adopt ViT-L/14 as the vision encoder and explore three different temporal modeling modules to learn multifaceted features for enhanced video understanding. In addition, we implement a two-phase training approach for Valley: the first phase focuses solely on training the projection module to facilitate the LLM's capacity to understand visual input, and the second phase jointly trains the projection module and the LLM to improve their instruction following ability. Extensive experiments demonstrate that Valley has the potential to serve as an effective video assistant, simplifying complex video-understanding scenarios. Our code and data are published anonymously at `https://github.com/valley-vl/Valley`.

## 1 Introduction

Large Language Models (LLMs) such as GPT (Ouyang et al., 2022), PaLM (Chowdhery et al., 2023), and LLaMA (Touvron et al., 2023) have demonstrated an exceptional ability to understand and follow user intentions and instructions. LLMs can learn knowledge from large-scale text corpora and have demonstrated remarkable performance across various language tasks, including text generation, summarization, machine translation, question answering, etc. In addition, one of the most significant changes brought by LLMs is the ability to handle conversational interactions as humans. By enabling natural and intuitive conversations, LLMs have paved the way for smoother human-computer interactions. A distinguished example of such work is ChatGPT(OpenAI, 2022), which has become an indispensable aspect of various applications, including customer service, healthcare, and e-commerce, commonly serving as AI assistants.

With the tremendous success of LLMs in language tasks, a natural question is: *Can we leverage them to better fuse visual and textual modalities and create multi-modal AI assistants?* Recent studies have achieved significant progress in this direction, especially in the area of image understanding, partially owing to the abundance and accessibility of publicly available image-textual data. We have witnessed the emergence of models such as InstructBLIP (Dai et al., 2023), Otter (Li et al., 2023a), Mini-GPT4 (Zhu et al., 2024), and LLaVA (Liu et al., 2023). In terms of architecture, these models conventionally feature a pre-trained vision encoder, an LLM, and a vision-language connector, e.g., Q-former (Li et al., 2023b) and MLP projection (Liu et al., 2023), to align visual and textual modalities. The advancement of video LLMs, however, has been much more challenging. Despite the ubiquity of video data, constructing large and high-quality video-textual and instruction datasets is a highly demanding task. VideoChat (Li et al., 2023c) first introduced a video-centric multi-modal instruction fine-tuning dataset. Then, Video-ChatGPT (Maaz et al., 2024) constructed the VideoInstruct100K dataset that was widely used in subsequent studies (Li et al., 2024; Wang et al., 2023). However, these video datasets suffer from poor quality, lack of diversity in content, and insufficient variety

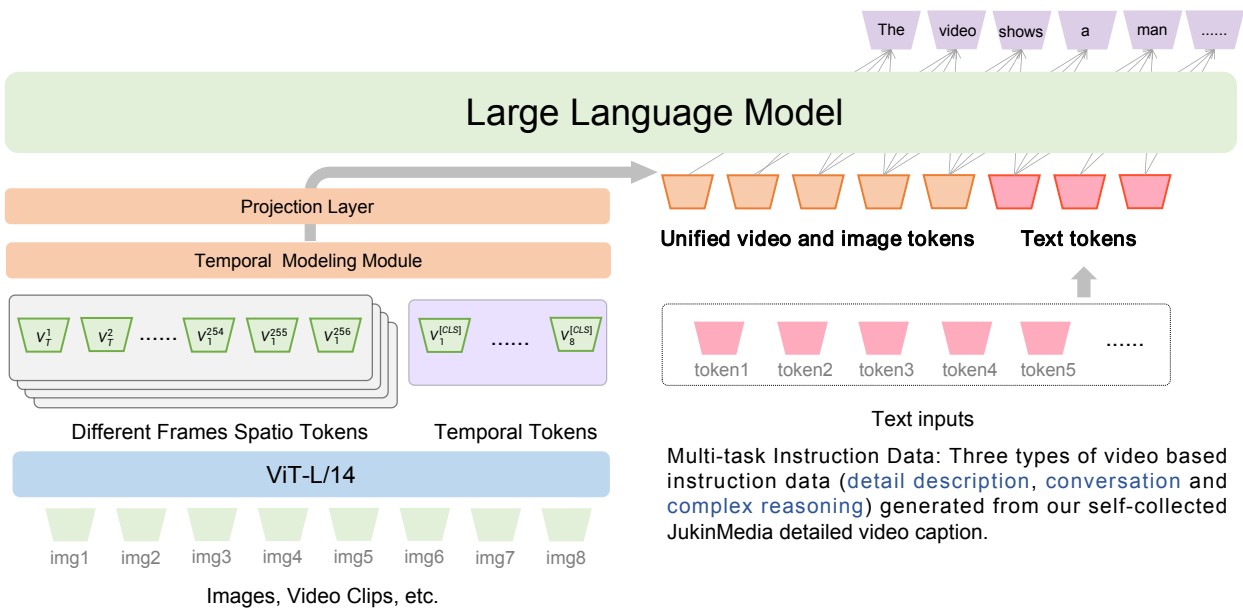

Figure 1: Illustration of the architecture of Valley.

in visual comprehension tasks. Moreover, temporal features in the video data have not been adequately excavated in previous models.

To address these issues, in this paper, we propose a novel multi-modal foundation model, *Valley*, which is capable of comprehending video and text in a unified framework, as illustrated in Figure 1. For the training of Valley, we first collect about 100,000 video samples and organize them into two video datasets with the assistance of ChatGPT, namely '*Valley-702k*' for video-text alignment and '*Valley-instruct-73k*' for video instruction tuning. The datasets encompass a diverse range of video-based instruction tasks, including multi-shot captions, long video descriptions, action recognition, causal inference, etc. The quality of both datasets is further enhanced by filtering out erroneous object information resulting from flawed predictions of vision models. To improve the visual comprehension capacity of Valley, we adopt ViT-L/14 (Dosovitskiy et al., 2021) as the vision encoder and explore three different strategies in the temporal modeling module to generate unified visual tokens for LLMs. In addition, we adopt a two-stage training strategy for Valley: In the first (pre-training) stage, we exclusively train the projection module to enable the LLM to comprehend visual data; In the second (end-to-end training) stage, we train the projection module and the LLM together, ensuring that Valley responds aptly in accordance with the instructions.

Finally, extensive experiments on video-based question-answering and caption benchmarks demonstrate that Valley achieves good performance and powerful zero-shot capability. Valley achieves state-of-the-art performance on video question-answering tasks in the MSVD, MSRVTT, and ActivityNet datasets in a zero-shot manner. In addition, Valley also provides leading results in video-based text generation tasks. In a recent Video-Bench benchmark (Ning et al., 2023) for video understanding, Valley outperforms all competing methods in all three tasks (Video-Exclusive, Prior-knowledge, and Decision-Making). We also demonstrated the chain-of-thought and few-shot capabilities of Valley. In the Science-QA dataset, Valley exhibits its one-shot inference ability and performs slightly better than its zero-shot counterpart. When using chain-of-thought (by adding "you need to think step by step" to the system prompt), the performance of Valley is generally comparable to or even better than GPT-3.5.

**Paper Organization.** In the remainder of this paper, we first discuss the related work in Section 2. Then, we describe the dataset construction process in Section 3. The procedure for training the Valley model is presented in Section 4. The experimental setup and results are shown in Section 5. Finally, we conclude the entire paper in Section 6. Some additional experiments and case studies are left to the appendix.

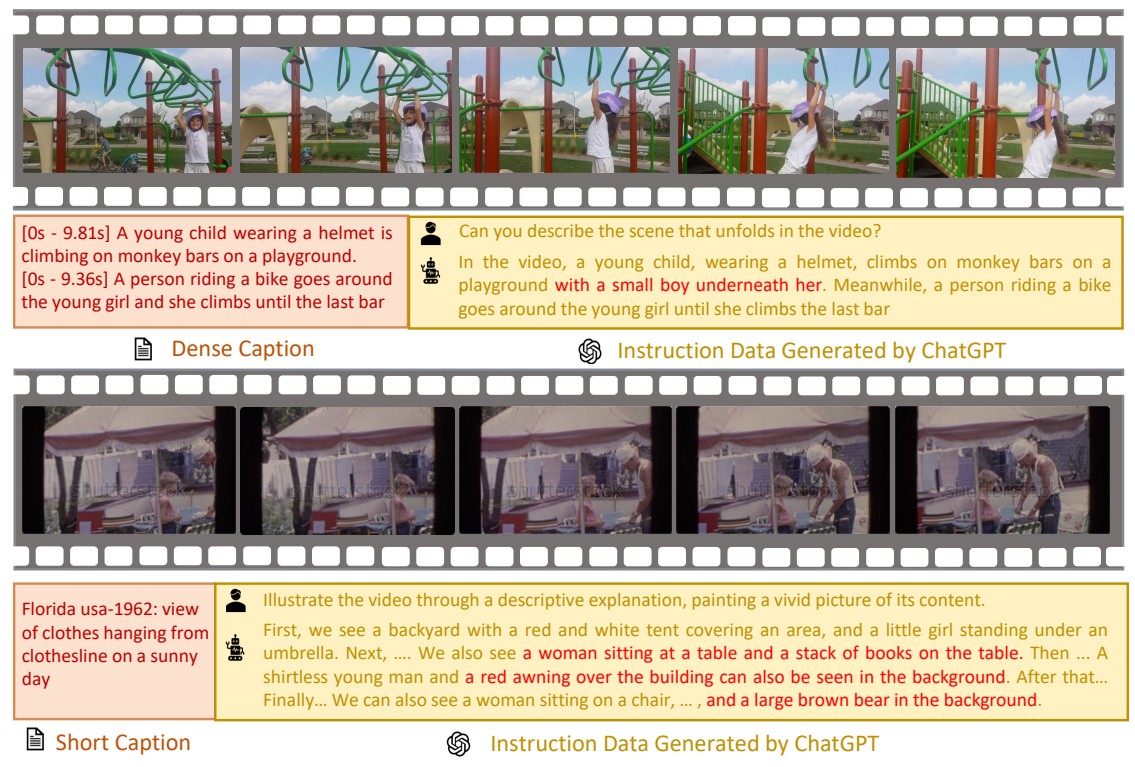

Figure 2: Two examples of hallucinations in the instruction data generated by dense captions (Video-ChatGPT) and short captions (VideoChat), where the illusion parts are highlighted in red.

## 2    Related Work

**Large Language Models (LLMs).**    LLMs have gained great success in natural language processing (Chowdhery et al., 2023; Ouyang et al., 2022; Hoffmann et al., 2022) due to their excellent language understanding and reasoning capacity. LLMs can handle complex linguistic tasks by comprehending prompts in a few-shot or zero-shot manner, and thus have been used in many different applications. Further, the development of a series of open-source LLMs, including LLaMA (Touvron et al., 2023), GLM (Du et al., 2022), and BLOOM (Scao et al., 2022), has inspired several AI assistants, such as Alpaca (Taori et al., 2023), Vicuna (Chiang et al., 2023), and ChatGLM Zeng et al. (2023). Due to their huge number of parameters, LLMs not only gain notable task transfer generalization ability but also complete actual tasks conversationally by aligning with human instructions and preferences. Inspired by these efforts, we aim to further extend LLMs to video-grounded conversation scenarios.

**Multimodal LLMs.**    As LLMs have demonstrated strong general capabilities in linguistic tasks, enabling LLMs to understand multimodal content has attracted increasing attention recently. Existing methods for multimodal LLMs can be divided into two categories based on their main techniques: one is to employ an LLM as a scheduler to organize the existing multimodal models, and the other is to train a multimodal model based on LLMs. After receiving user instructions and the functions of each foundation model, the former type of method treats the LLM as a controller to invoke the corresponding models step by step and integrates the output of each model to generate results (Wu et al., 2023; Shen et al., 2023; Yang et al., 2023). For example, HuggingGPT (Shen et al., 2023) utilizes ChatGPT to select appropriate models in Hugging Face[1] according to their function description and summarizes their output results. The latter type of method equips an LLM with auxiliary modules to help them understand multimodal content through end-to-end training (Li et al., 2023c; Zhu et al., 2024; Zhang et al., 2023; Liu et al., 2023; Su et al., 2023; Dai et al.,

---

[1] https://huggingface.co/models

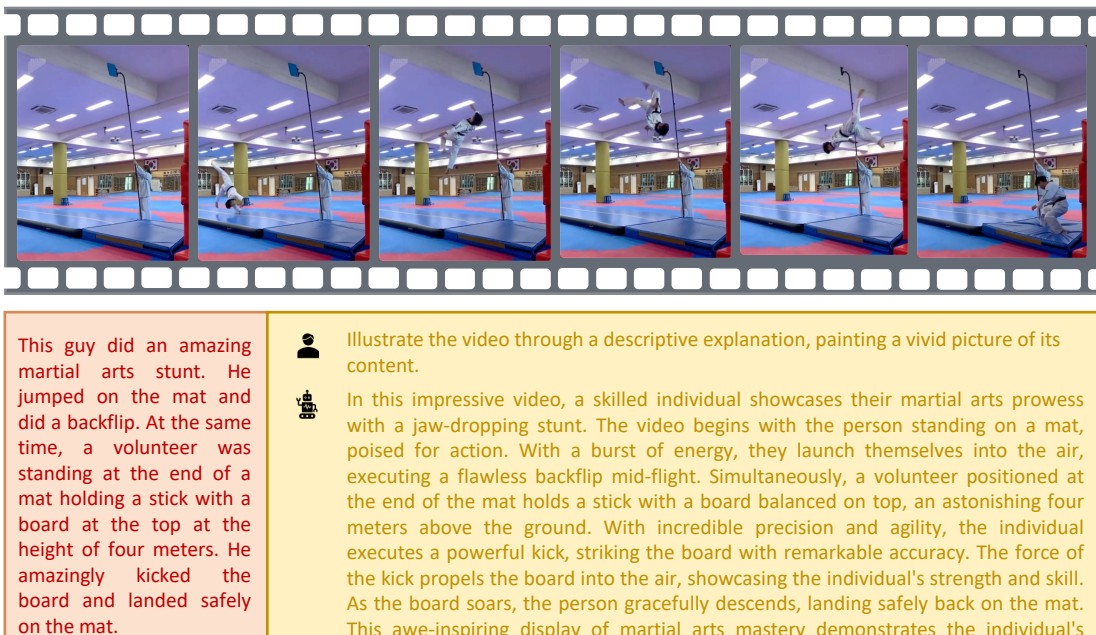

This guy did an amazing martial arts stunt. He jumped on the mat and did a backflip. At the same time, a volunteer was standing at the end of a mat holding a stick with a board at the top at the height of four meters. He amazingly kicked the board and landed safely on the mat.

Illustrate the video through a descriptive explanation, painting a vivid picture of its content.

In this impressive video, a skilled individual showcases their martial arts prowess with a jaw-dropping stunt. The video begins with the person standing on a mat, poised for action. With a burst of energy, they launch themselves into the air, executing a flawless backflip mid-flight. Simultaneously, a volunteer positioned at the end of the mat holds a stick with a board balanced on top, an astonishing four meters above the ground. With incredible precision and agility, the individual executes a powerful kick, striking the board with remarkable accuracy. The force of the kick propels the board into the air, showcasing the individual's strength and skill. As the board soars, the person gracefully descends, landing safely back on the mat. This awe-inspiring display of martial arts mastery demonstrates the individual's exceptional athleticism, timing, and control.

📄 Detailed Caption                    Instruction Data Generated by ChatGPT

Figure 3: An example of the instruction data generated from our collected detailed caption.

2023). For example, LLaVA (Liu et al., 2023) and MiniGPT-4 (Zhu et al., 2024) connect LLaMA (Touvron et al., 2023) with a visual encoder through a projection layer, endowing it with the ability to understand images. Video-LLaMA (Zhang et al., 2023) empowers LLaMA (Touvron et al., 2023) with both visual and audio information via Q-Former to endow it with video-grounded conversation ability.

## 3  Dataset Construction

There have been several studies on collecting video-based instruction data, such as VideoChat (Li et al., 2023c), Video-ChatGPT (Maaz et al., 2024). VideoChat and Video-ChatGPT use Webvid (Bain et al., 2021) and ActivityNet (Fabian Caba Heilbron & Niebles, 2015) to build their instruction dataset, respectively. Both of them suffer from some drawbacks when used for instruction data generation. The video captions generated by Webvid are often too short to contain sufficient information for ChatGPT to generate high-quality instruction data, whereas the video captions constructed by ActivityNet focus too much on human activities but often ignore other necessary information. To address these issues, they also use dense object caption generation methods such as GRIT (Wu et al., 2024) and Tag2Text (Huang et al., 2024) to provide object information in the video when constructing the instruction data. However, this will reduce the quality of the generated data, leading to strong hallucinations in the trained multimodal LLM. In Figure 2, we present two typical illusion examples of the instruction data generated by VideoChat and Video-ChatGPT.

To address the limitations of existing video datasets, we constructed the '*Valley-702k*' and '*Valley-instruct-73k*' datasets: the prior for aligning visual and textual modalities and the latter for improving the instruction-following capabilities. In order to endow Valley with the ability to understand visual information, we filter the WebVid2M (Bain et al., 2021) dataset using the same approach as LLaVA, We utilized the SpaCy library to perform phrase extraction from the video captions in the WebVid2M dataset and subsequently calculated their frequencies of occurrence. Based on this analysis, we constructed a candidate set of phrases, retaining only those with a frequency greater than 5. This threshold was chosen to exclude phrases with extremely low frequency, as their semantic information is likely subsumed by other higher-frequency phrases. The

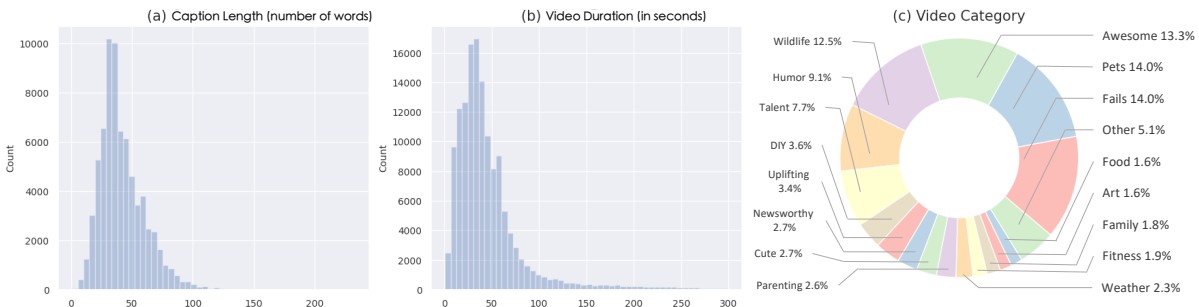

Figure 4: Distribution of video description length, duration, and type in the Valley dataset of about 100k videos collected from the JukinMedia website.

candidate phrases were then sorted in ascending order of frequency. For each phrase, we added captions containing the phrase to the dataset. If the number of captions containing a specific phrase exceeded 100, we randomly sampled 100 captions for inclusion. This process resulted in the construction of a new dataset for video-text alignment, referred to as *Valley-702k*. The selected '*Valley-702k*' dataset contains about 702,000 multi-turn dialogues using various questions to inquire about the content of the video and answer these questions using the corresponding captions. For instruction tuning, we first collect around 100,000 videos derived from JukinMedia, a website that provides diverse videos with high-quality descriptions. The dataset consists of video descriptions averaging 40 words, some exceeding 100 words, and videos with an average duration of 40 seconds, some longer than 5 minutes, providing sufficient temporal context. Figure 4 shows the distribution of video description length, video duration, and category among the videos that we collect. Then, we refer to the prompts for the generation of instruction data in LLaVA (Liu et al., 2023) and design three different types of prompts (for detail description, conversation, and complex reasoning) to build our instruction dataset based on these detailed video-text pairs. Given these prompts, we provide a manually written example and let ChatGPT generate it in a few-shot manner. Finally, we construct the '*Valley-instruct-73k*' dataset that contains 37k dialogue pairs, 26k complex reasoning QA pairs, and 10k detailed description instruction pairs for instruction tuning.

## 4 The Valley Model

In this section, we first introduce the architecture of our proposed Valley model. Then, we present the three structural designs of the temporal modeling module in Valley. Finally, we describe the two-stage training procedure of Valley.

### 4.1 Architecture

To enable a pre-trained large language model (LLM) to comprehend videos and jointly process videos of varying lengths along with individual images, we incorporate a temporal modeling module into the vision encoder. This module aggregates the grid features from each video frame into unified vision tokens. The overall architecture is shown in Figure 1.

We input a video $\boldsymbol{v}$ and sample $T$ frames by 0.5 FPS (1 picture per 2 seconds), which can be denoted as $\boldsymbol{v} = [v_1, v_2, \ldots, v_T]$. Each image obtains visual features through the pre-trained CLIP visual encoder (ViT-L/14), denoted as $\boldsymbol{V}_T = \text{ViT}(\boldsymbol{v}_T)$. Each feature contains 256 spatial patch features and 1 global feature ("[CLS]" token), i.e.,

$$\boldsymbol{V}_T = [\boldsymbol{V}_T^{[\text{CLS}]}, \boldsymbol{V}_T^1, \boldsymbol{V}_T^2, \ldots, \boldsymbol{V}_T^{256}].$$

We use a temporal modeling module (Section 4.2) to aggregate spatial patch features of $T$ frames in the time dimension, i.e.,

$$\widehat{\boldsymbol{V}} = \text{TEMPORALMOUDULE}\big([\boldsymbol{V}_1, \boldsymbol{V}_2, \ldots, \boldsymbol{V}_T]\big).$$

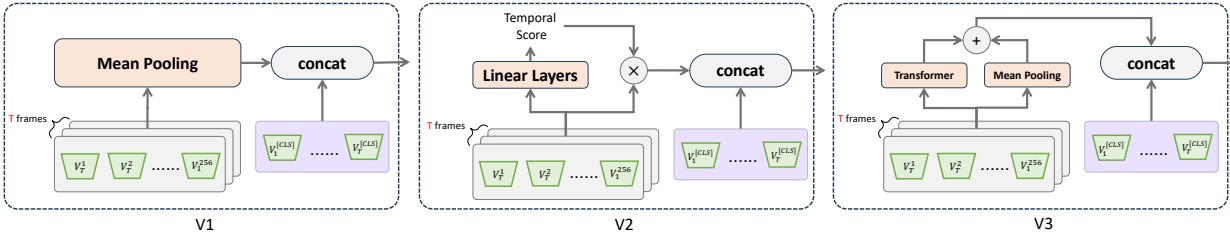

Figure 5: An illustration of the three types of temporal modeling modules in Valley.

To alleviate the vanishing of global temporal features caused by spatial feature aggregating, we obtain the representation $\boldsymbol{Z}_V$ of the entire video by concatenating patch features after the temporal modeling module and global features of $T$ frames, which is expressed as:

$$\boldsymbol{Z}_V = [\widehat{\boldsymbol{V}} \oplus \boldsymbol{V}_1^{[\text{CLS}]} \oplus \boldsymbol{V}_2^{[\text{CLS}]} \oplus \cdots \oplus \boldsymbol{V}_t^{[\text{CLS}]}],$$

where $\oplus$ is the concatenation operation.

We also adopt a projection layer to connect visual features into the word embedding space for its effectiveness in LLaVA. We utilize a trainable matrix to transform the video patch features and global features into a unified language feature space of the same dimension. Finally, the projected visual features and text embedding are input into an LLM for response generation. Formally,

$$\widehat{\boldsymbol{Z}_V} = \text{LLM}\big(\text{Projector}(\boldsymbol{Z}_V)\big).$$

## 4.2 Temporal Modeling Module

We propose three different structures (v1, v2, and v3) to aggregate the representation of spatial tokens in the time dimension, as shown in Figure 5.

In the v1 structure, we use an average pooling layer to aggregate spatial token representations:

$$\widehat{\boldsymbol{V}^i} = \text{AvgPooling}\big([\boldsymbol{V}_1^i, \boldsymbol{V}_2^i, \ldots, \boldsymbol{V}_T^i]\big),$$

where $i$ is the spatial token index. However, the v1 structure will lead to confusion in the time dimension as it does not consider that features in different timestamps may not be equally important.

The proposed v2 structure employs a learnable linear layer to assign temporal importance scores to each video frame, enabling a weighted averaging that distinguishes the relative importance of different frames:

$$\widehat{\boldsymbol{V}^i} = \text{Linear}\big([\boldsymbol{V}_1^i, \boldsymbol{V}_2^i, \ldots, \boldsymbol{V}_T^i]\big) \cdot [\boldsymbol{V}_1^i, \boldsymbol{V}_2^i, \ldots, \boldsymbol{V}_T^i].$$

In addition, we propose the v3 structure to model the temporal variation of spatial tokens.

We input spatial tokens into a one-layer transformer encoder and take the representation of the last timestamp output as the variation feature of the spatial token in temporal information. Then, we add the feature representing temporal information to the average pooling feature. The v3 formulation is as follows:

$$\widehat{\boldsymbol{V}^i} = \text{Transformer}\big([\boldsymbol{V}_1^i, \boldsymbol{V}_2^i, \ldots, \boldsymbol{V}_T^i]\big) + \text{AvgPooling}\big([\boldsymbol{V}_1^i, \boldsymbol{V}_2^i, \ldots, \boldsymbol{V}_T^i]\big).$$

# 5 Experiments

## 5.1 Setup

**Training Procedure.** Inspired by LLaVA, we adopt a two-stage training scheme. The first stage pre-trains the projection layer for feature alignment; and the second stage fine-tunes the language model and

projection layer jointly. Valley supports the input of any number of images, so in the pre-training phase, we use image-text pairs and video-text pairs for pre-training. The pre-training data includes 595K CC3M image-text pairs provided by LLaVA and 702K WebVid2M (Bain et al., 2021) video-text pairs filtered using the same method as in LLaVA. Both images and videos are input into the model in a unified way, and the prompt is as follows:

$$### \ X_{\text{system message}}(\text{e.g., you are an intelligent assistant} \ldots)$$
$$### \ \text{Human:} \ X_{\text{instruction}} \ \langle p_1 \rangle \ \ldots \ \langle p_{256} \rangle \ \langle f_1 \rangle \ \ldots \ \langle f_T \rangle$$
$$### \ \text{Assistant:}$$

If a single image is input, the number of frames is 1. The image-text pair and the video-text pair are constructed as a single-round dialogue, using various questions to inquire about the video content and answering with the corresponding caption.

As introduced in Section 3, we construct a 73k video-based instruction dataset that consists of 37k conversation pairs, 26k complex reasoning QA pairs, and 10k detail description instruction pairs. In order to enhance the ability to describe visual content in detail, we also collect 150k image instruction data from LLaVA and 11K video instruction data from VideoChat. We use the total 234k video- and image-based instruction data to perform fine-tuning in the second stage, which freezes the ViT weight and adjusts all parameters in the projection layer and the LLM jointly.

**Datasets and Tasks.** We evaluate Valley on six diverse datasets across two domains: (1) MSVD-QA (Chen & Dolan, 2011), MSRVTT-QA (Xu et al., 2016), ActivityNet-QA (Yu et al., 2019), and Video-ChatGPT (Maaz et al., 2024) for video understanding; (2) ScienceQA (Lu et al., 2022) and MemeCap (Hwang & Shwartz, 2023) for image understanding. All experiments were carried out in zero/few-shot manners.

For video question answering (MSVD-QA, MSRVTT-QA, and ActivityNet-QA), we use the prediction accuracy of all models and the rating scores of their outputs on a 5-point scale using ChatGPT (the prompt for evaluation is given in the technical appendix) as performance measures. For the Video-ChatGPT benchmark, we also score the outputs on the same 5-point scale using ChatGPT to evaluate their generation capabilities for video description and question answering. On these four datasets, we compare Valley with FrozenBiLM (Yang et al., 2022), VideoChat (Li et al., 2023c), LLaMA Adapter (Zhang et al., 2024), Video-LLaMA (Zhang et al., 2023), and Video-ChatGPT (Maaz et al., 2024).

For ScienceQA, a question-answering dataset on natural, social, and language science in elementary and high school science curricula, we assemble the textual and image context of each question as input and adopt accuracy as the performance measure. For MemeCap, a content analysis dataset that requires the model to generate a description based on the title and images of posts, we use the BERT F1-score, which reflects the semantic similarity between the generated text and the ground truth, for evaluation. On these two datasets, we compare Valley with Flamingo (Alayrac et al., 2022) and MiniGPT4 (Zhu et al., 2024). We also use GPT-3.5 (Lu et al., 2022) as a baseline on ScienceQA.

**Implementation Details.** In the Valley implementation, we used Stable-Vicuna (Chiang et al., 2023) as the LLM backbone and the pre-trained ViT-L/14 (Dosovitskiy et al., 2021) to encode videos and images. We first pre-trained Valley for one epoch with a learning rate of $2 \times 10^{-3}$ and then fine-tuned the model for three epochs with a learning rate of $2 \times 10^{-5}$ on the instruction dataset. All experiments were carried out on 8 Nvidia A100 80G GPUs.

## 5.2 Experimental Results

**Zero-Shot Video Question-Answering.** Table 1 shows the results of zero-shot inference on three video question-answering datasets. For Valley, we report its performance with three different temporal modeling structures. Generally, Valley shows remarkable superiority over most baseline models. Among its different versions, Valley-v1 outperforms a significant number of baseline models, while Valley-v3 stands out as the top performer. It demonstrates that Valley-v3 has an outstanding ability to understand the video context and can offer more accurate and reasonable answers compared to other models.

Table 1: Results for zero-shot video question-answering.

| Model | MSVD | | MSRVTT | | ActivityNet | |
|---|---|---|---|---|---|---|
| | Acc. | Score | Acc. | Score | Acc. | Score |
| FrozenBiLM | 32.2 | – | 16.8 | – | 24.7 | – |
| VideoChat | 56.3 | 2.8 | 45.0 | 2.5 | 26.5 | 2.2 |
| LLaMA Adapter | 54.9 | 3.1 | 43.8 | 2.7 | 34.2 | 2.7 |
| Video-LLaMA | 51.6 | 2.5 | 29.6 | 1.8 | 12.4 | 1.1 |
| Video-ChatGPT | 64.9 | 3.3 | 49.3 | 2.8 | 35.2 | 2.7 |
| Grounding-GPT | 67.8 | 3.7 | **51.6** | 3.1 | 44.7 | 3.2 |
| Valley-v1 | 65.4 | 3.4 | 45.7 | 2.5 | 42.9 | 3.0 |
| Valley-v2 | 59.1 | 3.4 | 49.9 | 3.0 | 32.5 | 2.6 |
| Valley-v3 | **69.2** | **4.0** | 50.8 | **3.3** | **44.9** | **3.4** |

**Video-based Text Generation.** To verify the text generation performance of each model, we evaluated their generated texts using ChatGPT in five aspects, namely correctness (**COR**), detail orientation (**DO**), contextual understanding (**CU**), temporal understanding (**TU**), and consistency (**CON**). The experimental results on the Video-ChatGPT benchmark are shown in Table 2. We find that Valley-v3 achieves the best performance in four out of five aspects (COR, CU, TU, CON). This is consistent with the results in Table 1, as the Video-ChatGPT benchmark is built on top of ActivityNet videos, and Valley-v3 has been shown to perform well in understanding longer videos. We speculate that the mean pooling in Valley-v1 is suboptimal for fully understanding the details and consistency of the video, while Valley-v3 benefits from the refined temporal modeling module and achieves better performance. From the perspective of suppressing hallucinations, the dataset we construct can make Valley perform better in terms of correctness. In the aspect of detail orientation, Valley is slightly inferior to Video-ChatGPT. This is because the instruction data we construct misses some detailed descriptions of objects in order to ensure correctness.

Table 2: Scores for video-based text generation tasks in the Video-ChatGPT benchmark.

| Model | COR | DO | CU | TU | CON |
|---|---|---|---|---|---|
| VideoChat | 2.23 | 2.50 | 2.53 | 1.94 | 2.24 |
| LLaMA Adapter | 2.03 | 2.32 | 2.30 | 1.98 | 2.15 |
| Video-LLaMA | 1.96 | 2.18 | 2.16 | 1.82 | 1.79 |
| Video-ChatGPT | 2.40 | **2.52** | 2.62 | 1.98 | 2.37 |
| Valley-v1 | 2.06 | 2.42 | 2.74 | 1.83 | 2.41 |
| Valley-v2 | 2.35 | 2.13 | 2.85 | 1.99 | 2.17 |
| Valley-v3 | **2.43** | 2.13 | **2.86** | **2.04** | **2.45** |

**Video-Bench Tasks.** As shown in Table 3, Valley outperforms all competing methods in all three tasks (Video-Exclusive, Prior-knowledge, and Decision-Making) of the Video-Bench (Ning et al., 2023) benchmark. This further confirms the effectiveness of Valley in video-based tasks.

Table 3: Results for tasks in the Video-Bench benchmark.

| Model | All | Video-Exclusive | Prior-Knowledge | Decision-Making |
|---|---|---|---|---|
| VideoChat | 35.4 | 34.1 | 29.6 | 42.5 |
| Video-LLaMA | 32.8 | 32.5 | 27.8 | 38.2 |
| Video-ChatGPT | 38.5 | 39.8 | 29.2 | 46.5 |
| Valley-v3 | **41.3** | **40.4** | **35.8** | **47.7** |

**Image & Textual Understanding.** In addition to video understanding, we also evaluate our Valley model on several main image and textual understanding tasks. In terms of image metaphor understanding, the texts generated by Valley are semantically closer to the ground truth than those generated by MiniGPT4 and Flamingo, as shown in Table 4.

Table 4: Results on the MemeCap dataset for the image metaphor understanding task.

| Model | Setup | BERT F1-score |
|---|---|---|
| Flamingo | zero-shot | 65.51 |
| | zero-shot COT | 58.23 |
| MiniGPT4 | zero-shot | 65.81 |
| | zero-shot COT | 68.45 |
| Valley | zero-shot | 84.82 |
| | zero-shot COT | **85.26** |

We also explore the chain-of-thought and few-shot capabilities of Valley on the Science-QA dataset. From the experimental results in Table 5, we observe that Valley exhibits a certain one-shot inference ability and performs slightly better than its zero-shot counterpart. We also notice that when using chain-of-thought (add "you need to think step by step" to the system prompt), the performance of Valley is generally comparable to GPT-3.5 and is even better than GPT-3.5 in a few aspects, such as G7-12, SOC, and TXT, while the latter has much more parameters than the former.

Table 5: The accuracy (%) for each question class on the ScienceQA dataset. Question classes: NAT for natural science, SOC for social science, LAN for language science; TXT for text context, IMG for image context, NO for no context; G1-6 for grades 1-6, G7-12 for grades 7-12.

| Model | Subject | | | Context Modality | | | Grade | | Average |
|---|---|---|---|---|---|---|---|---|---|
| | NAT | SOC | LAN | TXT | IMG | NO | G1-6 | G7-12 | |
| (Human) | 90.23 | 84.97 | 87.48 | 89.60 | 87.50 | 88.10 | 91.59 | 82.42 | 88.40 |
| GPT-3.5 | 74.64 | 69.74 | 76.00 | 74.44 | 67.28 | 77.42 | 76.80 | 68.89 | 73.97 |
| Valley (Zero-Shot) | 69.89 | 71.54 | 69.90 | 73.00 | 67.22 | 72.22 | 72.32 | 66.51 | 70.24 |
| Valley (Zero-Shot + COT) | 71.00 | 72.32 | 74.72 | 76.90 | 66.98 | 80.55 | 73.49 | 70.01 | 72.25 |
| Valley (One-Shot) | 71.53 | 63.66 | 74.54 | 75.83 | 64.70 | 83.33 | 71.51 | 69.14 | 70.66 |

**Case Studies.** We present several practical cases to demonstrate the superior video understanding and instruction-following capabilities of Valley.

As shown in Figures 6–8, the text generated by Valley based on videos demonstrates several advantages. First, the text provides an accurate description of the relationship. In Figure 6, it accurately identifies the objects that interact with the cat. In Figure 7, it highlights the strong bond between the person and the dog, highlighting their connection and mutual enjoyment of spending time together. Second, the text captures the emotional tone. In Figure 6, it finds the funny point within the video. In Figure 7, it effectively captures the playful nature of the interaction between the person and the dog, reflecting the joy and companionship they share. Third, the text gives an imaginative interpretation. In all figures, it provides a creative and engaging narrative, particularly in the short story section of Figure 7 and the cooking action description section of Figure 8. Fourth, the text contains detailed observations. In Figure 7, it offers detailed observations about the dog's actions, noting its ability to wear clothes and interact with the computer, which contributes to a vivid description of the scene. In Figure 8, it corrects the errors in a given description of the video.

Overall, the text successfully conveys the essence of the video and adds a layer of imaginative interpretation, making the description engaging and emotionally resonant.

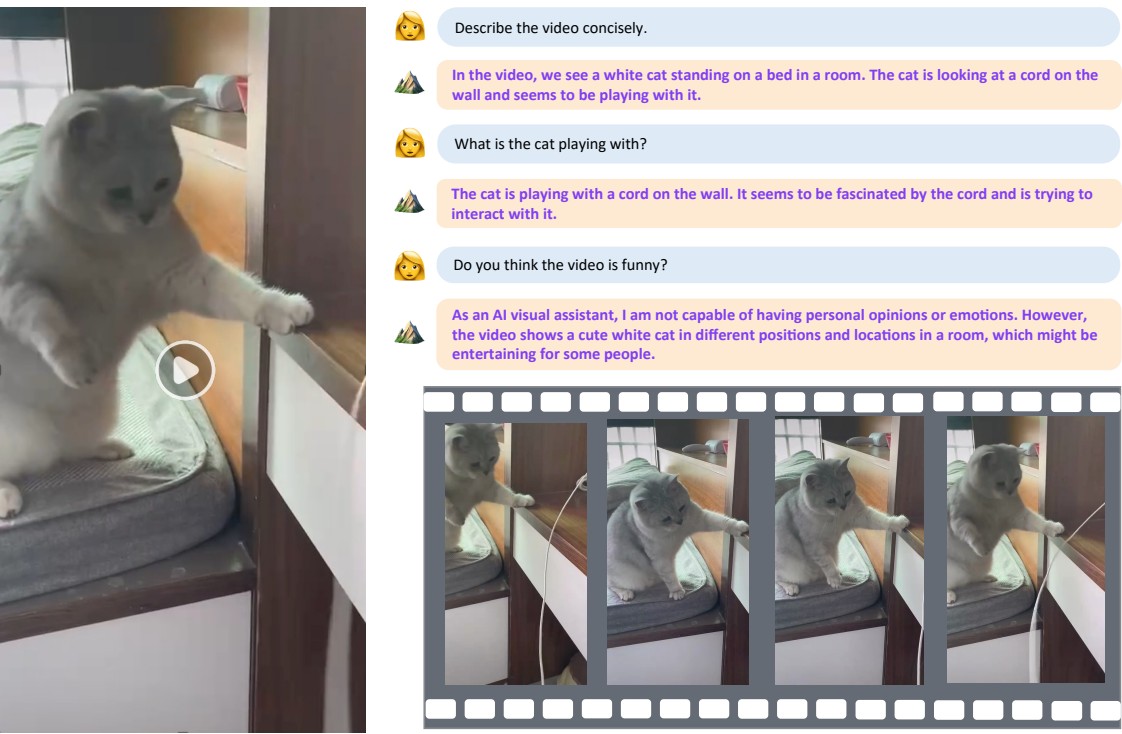

Figure 6: A case study of using Valley for question answering based on a video about a cat.

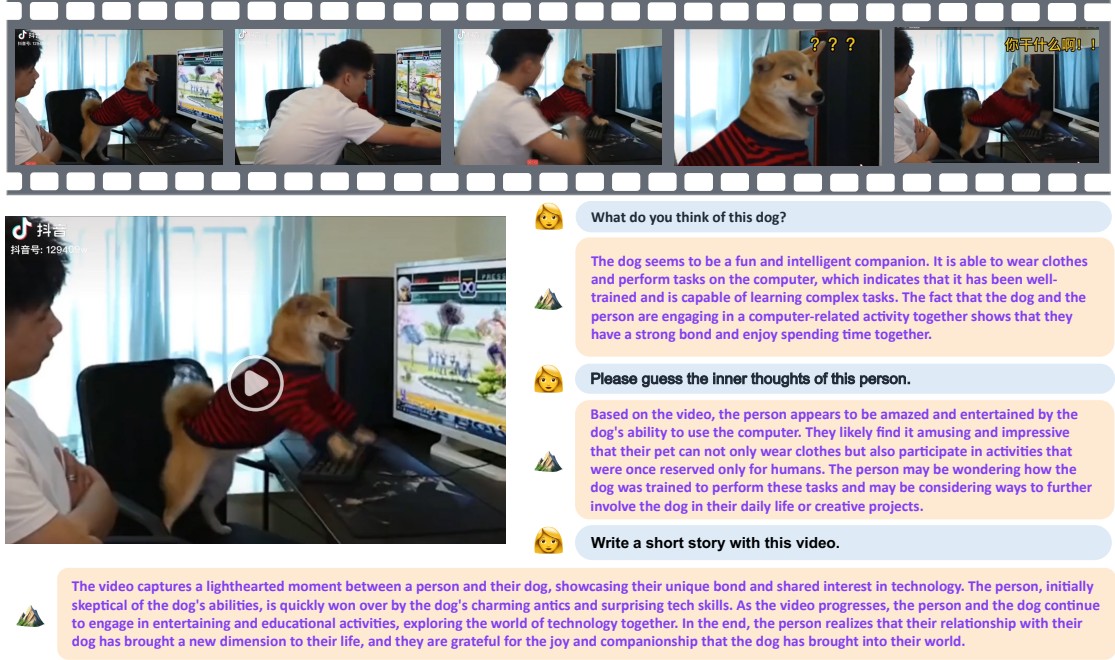

Figure 7: A case study of using Valley for question answering based on a video about the interactions of a man and his dog.

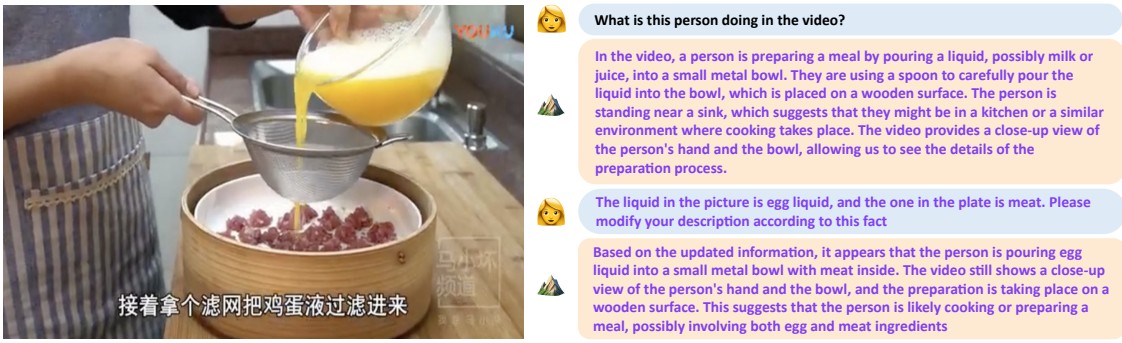

Figure 8: A case study of using Valley for question answering based on a video about cooking.

# 6 Conclusion

In this paper, we propose a multimodal foundation model, Valley, which is capable of understanding video, image, and language in a unified manner. We first collect 100k videos with detailed captions and construct a multi-task video instruction-following dataset of high quality with the assistance of ChatGPT for the training of Valley. Also, the video-text pairs from the Webvid dataset are carefully cleaned to improve data quality. We finally constructed two datasets, namely '*Valley-702k*' and '*Valley-instruct-73k*'. We further design three structures for the temporal modeling module and adopt a two-stage training procedure for Valley. Finally, extensive experiments on several multimodal datasets confirm the effectiveness of Valley in video, image, and textual understanding and generation. Our ultimate goal in future work is to create a more intuitive and personalized multimodal AI assistant to facilitate human-machine interactions.

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

Table 6: Implementation settings of Valley.

| Configuration | Stage 1 (Pre-training) | Stage 2 (Instruction Tuning) |
|---|---|---|
| Training dataset | '*Valley-702k*' | '*Valley-instruct-73k*' |
| ViT initialization | | Open-CLIP-L-224 |
| LLM initialization | Stable-Vicuna-13b | Valley after Stage 1 |
| Connection module initialization | random | Valley after Stage 1 |
| Image resolution | | $224 \times 224$ |
| ViT sequence length | | 256 |
| LLM sequence length | | $2,048$ |
| Optimizer | | AdamW |
| Optimizer hyperparameter | | $\beta_1 = 0.9, \beta_2 = 0.95, eps = 10^{-6}$ |
| Learning rate | $2 \times 10^{-3}$ | $5 \times 10^{-5}$ |
| Learning rate schedule | | cosine decay |
| Weight decay | | 0.0 |
| Gradient checkpointing | | ✓ |
| Training epoch | 1 | 3 |
| Warm-up ratio | | 0.03 |
| Global batch size | 128 | 32 |
| Numerical precision | | `bfloat16` |
| Optimizer sharding | | ✓ |
| Activation checkpointing | | ✗ |
| Model parallelism | | Zero2 |
| Pipeline parallelism | | ✗ |

## A   Implementation Details of Valley

We report the detailed implementation settings of Valley in Table 6.

The training process of Valley involves two stages, Stage 1 (Pre-training) and Stage 2 (Instruction Tuning), as indicated in the main paper. In the first pre-training stage, the Vision Transformer (ViT) is initialized with Open-CLIP-L-224, ensuring a robust visual backbone, while the Large Language Model (LLM) is initialized using Stable-Vicuna-13b. A random initialization is applied to the connection module, which bridges the visual and language components. The training uses images with a resolution of $224 \times 224$ pixels and a ViT sequence length of 256, paired with an LLM sequence length of 2,048. The optimizer used is AdamW with hyperparameters $\beta_1 = 0.9$, $\beta_2 = 0.95$, and $\epsilon = 10^{-6}$, and the learning rate is set to $2 \times 10^{-3}$ following a cosine decay schedule. The global batch size is 128, and training is carried out for one epoch with a warm-up ratio of 0.03, utilizing mixed precision with `bfloat16` to optimize computational efficiency. Additionally, the model leverages optimizer sharding and gradient checkpointing while implementing Zero2 model parallelism.

In the second stage, Instruction Tuning, the LLM and connection module are both initialized from the parameters obtained in the first stage, ensuring continuity in the training process. The learning rate is significantly reduced to $5 \times 10^{-5}$ to fine-tune the model with greater precision, while other key hyperparameters, such as the use of AdamW optimizer, cosine decay learning rate schedule, and `bfloat16` numerical precision, are kept the same as those in the pre-training phase. This stage increases the number of training epochs to three but reduces the global batch size to 32 to allow for more focused updates during the instruction tuning process. The use of optimizer sharding and gradient checkpointing is maintained, with the Zero2 model parallelism still in effect.

## B   Additional Case Studies

Figure 9 shows two cases about the capacity of Valley to understand image metaphors.

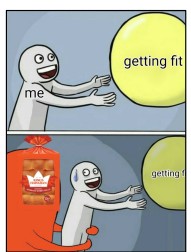

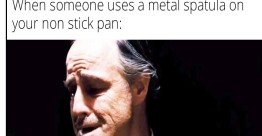

Figure 9: Two examples of Valley in the Meme-cap dataset. The blue box represents the human-annotated understanding of the image, and the green box represents the understanding of the image by Valley.

## C  Examples of Training Dataset

### C.1  Example of training data in '*Valley-702k*'

```
{
"id": "25116338",
"video": "037551_037600/25116338.mp4",
"conversations": [
    {"from": "human", "value": "Describe the following video concisely.\n<video>"},
    {"from": "gpt", "value": "Smiling young girl picking fingers on keyboard of personal
    computer message, make appointment or going to movies, sitting in chair sheltered
    plaid on background of white brick wall on background"}]
}
```

### C.2  Example of training data in '*Valley-instruct-73k*'

```
{
"id": "jukin_1285059",
"v_id": "1285059",
```

```
"video": "jukinmedia_v_1285059.mp4",
"source": "jukinmedia",
"conversations": [
    {"from": "human", "value": "<video>\nHow do the performers execute their mind-
    blowing flipping trick at the airport?"},
    {"from": "gpt", "value": "The performers execute their mind-blowing flipping trick
    at the airport by utilizing their acrobatic skills and the available space. They
    start by finding an open area in the airport, preferably with a smooth surface.
    One performer takes a running start and performs a powerful jump, propelling
    themselves into the air. As they reach the peak of their jump, they tuck their body
    into a tight ball and initiate a series of flips. Meanwhile, the other performer
    positions themselves strategically to catch the first performer mid-flip. This
    requires precise timing and coordination between the two performers. The second
    performer extends their arms and creates a stable platform for the first performer
    to land on. Once the first performer completes their flips, they extend their body
    and land securely on the second performer's outstretched arms. The second performer
    then uses their strength and balance to support the weight of the first performer,
    ensuring a safe landing. This impressive flipping trick showcases the performers'
    athleticism, coordination, and trust in each other's abilities."}]
}
```

## D  Prompt for Instruction Data Generation

The prompts we use to generate the three types of instruction data are shown in Tables 7, 8, and 9, respectively. Regarding the detailed description type and complex reasoning type data, we will give a manually written example and let ChatGPT generate it using few-shot learning.

## E  Prompt For Evaluating QA Task

The prompt for using ChatGPT to assist in the evaluation of the experiments is shown in Table 10. This prompt will allow the AI to judge the accuracy of the model prediction results and score the consistency of the answer.

Table 7: Prompt for generating instruction data of detail description with few-shot learning.

---

**Prompt for Detail Description**

**System**: You are an intelligent assistant that can understand video information through text descriptions. You can understand the overall content of the video from the title of the video, the caption of the video. Please describe the video you saw through the information given above. Don't mention the title in your description. Don't copy the original caption. Do not separately describe which objects are included in the video. It is necessary to integrate object information into your description through adjectives or attributive clauses. This description should be between 150 and 200 words.

**User**:
[title] Guy Scratches Head After Landing Perfect Bowling Strike [Caption] This guy scratched his head in confusion after making a mind-blowing attempt at bowling. He swung his hand to release the ball but accidentally tossed it towards the gutter. However, it spun and turned at the side edges of the lane and then struck all pins in one go.

**ChatGPT**: In the video, we see a man wearing a maroon shirt and shorts standing in a bowling alley, holding a bowling ball. First, he swings his hand to release the ball but accidentally tosses it towards the gutter. Next, the ball spins and turns at the side edges of the lane, seemingly heading towards the gutter, but suddenly changes direction and heads towards the pins.···

**User**:
[title] ··· [Caption] ···

---

Table 8: Prompt for generating instruction data of conversation.

**Prompt for Conversation**

**System**: The task is to generate a conversation between two people. One person is watching at a video, and the other person is asking questions about the video. What they see will be provided below with some sentences. Include at least one complex question that requires reasoning and thinking. Only include the questions with certain answers, then one can answer with the provided sentences. Make the QA sound like they are seeing the video. Do not use any words that may sound like looking at text instead of images, like "specify", "mention", "description", "text", "provided information", "sentence", "caption", etc. Use words like "see", "look", "view", "show", etc. Format each QA pair in a single line as a JSON dictionary. Do not include any other explanation.

**User**:
[title] · · · [Caption] · · ·

Table 9: Prompt for generating instruction data of complex reasoning with few-shot learning.

**Prompt for Detail Description**

**System**: You are an AI visual assistant that can analyze a single video. You receive a title of this video and a caption of this video, each describing the same video you are observing. The task is to use the provided title and caption, create a plausible question about the video, and provide the answer in detail.Create complex questions beyond describing the scene.To answer such questions, one should require first understanding the visual content, then based on background knowledge or reasoning, either explain why things are happening that way, or provide guides and help to user's request. Make the question challenging by not including the visual content details in the question so that the user needs to reason about that first. When using the information from the caption, directly explain the scene, and do not mention that the information source is the caption. Always answer as if you are directly looking at the video.

**User**:
[title] Woman Pranks Sister by Covering Inside of Her Whole House in Aluminium Foil
[Caption] This woman had gone on a vacation. However, she was shocked when she entered her house on returning. Her sister had covered her whole house with aluminum foil from inside to prank her. She laughed uncontrollably as she saw everything covered in the foil.

**ChatGPT**:
{
"question": "Given the sister's initial reaction of uncontrollable laughter upon discovering the prank, how might this prank affect their relationship in the long run, considering psychological and social aspects?",
"answer": " From a psychological perspective, humor plays a significant role in maintaining healthy relationships. The sister's reaction of laughter suggests that she found the prank amusing, which could enhance their bond. Shared laughter can increase feelings of intimacy and social cohesion, indicating that the prank may have strengthened their relationship. · · · "
}

**User**:
[title] · · · [Caption] · · ·

Table 10: Prompt for QA task evaluation following that of Video-ChatGPT.

**Prompt for Detail Description**
**System**: You are an intelligent chatbot designed for evaluating the correctness of generative outputs for question-answer pairs.
Your task is to compare the predicted answer with the correct answer and determine if they match meaningfully. Here is how you can accomplish the task:
− − − − − −
## INSTRUCTIONS:
- Focus on the meaningful match between the predicted answer and the correct answer.
- Consider synonyms or paraphrases as valid matches.
- Evaluate the correctness of the prediction compared to the answer.
**User**: Please evaluate the following video-based question-answer pair:

Question: {question}
Correct Answer: {answer}
Predicted Answer: {pred}

Provide your evaluation only as a yes/no and score where the score is an integer value between 0 and 5, with 5 indicating the highest meaningful match.
Please generate the response in the form of a Python dictionary string with keys 'pred' and 'score', where value of 'pred' is a string of 'yes' or 'no' and value of 'score' is in INTEGER, not STRING.
DO NOT PROVIDE ANY OTHER OUTPUT TEXT OR EXPLANATION. Only provide the Python dictionary string.
For example, your response should look like this: {'pred': 'yes', 'score': 4.8}."rom inside to prank her. She laughed uncontrollably as she saw everything covered in the foil.

