# OpenReview forum: "Valley: Video Assistant with Large Language Model Enhanced Ability"
_TMLR — Rejected by TMLR_

### Review · Reviewer_YG7V · 2024-12-14

**Summary Of Contributions:**

This paper introduces a multi-modal foundation model Valley, which comprehends video and text in a unified framework. The authors adopt a two-stage training strategy for training Valley. Specifically, they collect ‘Valley-702k’ for video-text alignment and ‘Valley-instruct-73k’ for video instruction tuning.  Valley achieves superior performance on video question-answering tasks and video-based text generation tasks.

**Audience:**

Yes

**Broader Impact Concerns:**

No ethical concerns.

**Claims And Evidence:**

Yes

**Requested Changes:**

From my perspective, this paper needs to adjust the following points, I think these points can help improve the paper:
1. Beyond accuracy and ChatGPT scores, the authors should employ additional metrics to evaluate model performance, such as BLEU, METEOR, and ROUGE, to provide a more comprehensive assessment of the quality of the generated text.
2. To assess the model's generalization ability, the authors should analyze the model's performance across various video types.
3. I suggest analyzing the advantages and disadvantages of different temporal modeling modules and explain why the v3 architecture performs the best.
4. I hope the authors can analyze the model's performance on videos of varying lengths to evaluate the sensitivity of the temporal modeling modules to video length.
5. The authors can add human evaluation to assess the quality of the instruction data generated by ChatGPT.

**Strengths And Weaknesses:**

The strength of this paper consists of:
1. The paper addresses limitations in existing video datasets like VideoChat and Video-ChatGPT, which suffer from shortcomings such as short captions or an overemphasis on human activities while neglecting other crucial information. To overcome these issues, the paper introduces two new datasets: "Valley-702k" and "Valley-instruct-73k."
2. To better capture the temporal characteristics of video data, the researchers incorporated temporal modeling modules within the visual encoder. They explored three different structural designs (v1, v2, and v3) to aggregate the spatial token representations across the time dimension. "Spatial token representations" refers to the model's internal representation of objects and features within each frame of the video.
3. This paper employs a two-stage training strategy that leverages both image and video data to achieve robust performance.

Meanwhile, the weakness contains:
1. The analysis of the temporal modeling module is insufficient. The paper proposed three temporal modeling modules but did not explain why the v3 structure performed best, and the specific shortcomings of other structures. Moreover, the paper did not analyze the performance on videos of different lengths, which is crucial for understanding the effect of the temporal modeling module.
2. Although the paper mentioned using ChatGPT to generate high-quality instruction data and filter the WebVid2M dataset, it lacks quantitative analysis of data quality. For example, the authors could calculate the proportion of erroneous object information in the dataset, or use human evaluation to judge the quality of the instruction data generated by ChatGPT. Furthermore, the paper did not discuss the impact of data bias. The JukinMedia and WebVid2M datasets may not be representative of all types of video content, which may cause the model to perform poorly in some cases.
3. The baseline of this paper is somewhat outdated, such as in Table 1 and Table 5, but I understand that recent models are really strong.

---

> ### Author Response · Authors · 2025-01-15
>
> We thank Reviewer YG7V for the comments that help improve this paper. Next, we provide a point-to-point response to each comment.
>
> **W1.** Regarding temporal module
>
> **Response:**  For improving v1, v2 introduces a learnable linear layer to learn the temporal importance score of a certain frame. In order to model the temporal variation representation of these spatial tokens more fully, we derived the v3 based on Transformer. In light of experiments, the performance of V2 is basically between V1 and V3, such as the MSRVTT-QA and the video-based text generation benchmark provided by Video-ChatGPT.
>
> **W2.** About more detailed and manual evaluations
>
> **Response:** We completely agree with your viewpoint and acknowledge the importance of high-quality data for our method. We are also well aware that video data in the domain is often sparse and requires a substantial amount of high-quality training data.  Therefore, in our current work, we have made significant efforts to gather the available high-quality data from open websites and design instructions and conversations to train the model. Based on our knowledge, it is of the largest scale in the existing video instruction dataset. And the results prove its "high quality".  However, we lack the manpower to conduct more meticulous data quality checks.  The impact of data bias definitely exists. The JukinMedia and WebVid2M datasets cannot represent all types of video content either. However, we have made initial efforts, hoping to contribute to the development of the community.

---

### Review · Reviewer_NDCQ · 2024-12-14

**Summary Of Contributions:**

The paper introduces a multimodal foundation model to enhance video comprehension and instruction-following capabilities. Valley incorporates a two-phase training strategy and leverages newly constructed datasets: Valley-702k for video-text alignment and Valley-instruct-73k for video-based instruction tasks. These datasets support diverse video-text interactions, including action recognition, causal inference, and long video descriptions. The model integrates ViT-L/14 as the vision encoder and explores three temporal modeling modules for spatiotemporal feature aggregation. The paper claims that Valley's performance in tasks like video question answering and video-text generation showcases zero-shot and chain-of-thought reasoning abilities.

**Audience:**

Yes

**Claims And Evidence:**

Yes

**Requested Changes:**

1. Please add more details about how you collect your dataset. Is it fully automatic, or are humans involved in the process? You mentioned, "We filter the WebVid2M (Bain et al., 2021) dataset using the same approach as LLaVA". Could you please briefly introduce them here?
2. The content of section 4.3 should be considered as part of your experiments rather than a part of your model introduction.
3. In Figure 9, please consider putting two examples horizontally. They waste too many spaces.

**Strengths And Weaknesses:**

Strengths:
1. The paper proposes innovative datasets. Valley-702k and Valley-instruct-73k cover a broad range of video-text tasks.
2. The model integrates advanced temporal modules to capture video dynamics better.
3. The approach ensures gradual skill acquisition, initially focusing on projection alignment and then joint LLM-instruction tuning.

Weaknesses:
1. The data construction process highly relies on high-quality datasets and lacks details and clarification of authors' works.
2. The proposed connection layer may not effectively handle long videos (such as those lasting more than 30 minutes). The transformer module in v3 may be time-consuming and cost-consuming when the number $V_T$ is large (>>256).
3. Section 4.3 is similar to the LLaVa training process. It doesn't seem like a novel method, which should not be described here.

---

> ### Author Response · Authors · 2025-01-15
>
> We thank Reviewer NDCQ for the comments that help improve this paper. Next, we provide a point-to-point response to each comment.
>
> **W1 & Q1.** Details of data construction
>
> **Response:** We have supplemented the corresponding data and implementation details in the revised manuscript. Additionally, we have published our code and data anonymously, which can further assist in reproducing our results.
>
> **W2.** Handling long videos
>
> **Response:** We should admit that our current model cannot adequately handle the scenarios of longer videos. In the subsequent versions of Vallay, we will consider how to support long videos.
>
> **W3 & Q2.** About Section 4.3
>
> **Response:** In the revised manuscript, we have moved Section 4.3 to Section 5.1 (Experimental Setup).
>
> **Q3.** About Figure 9
>
> **Response:** We now move Figure 9 to the appendix to avoid wasting too many spaces in the main paper.

---

### Review · Reviewer_5Umb · 2024-12-17

**Summary Of Contributions:**

This paper explores the joint video and language understanding in model training. The authors proposed Valley, a mujlti-modal foundation model designed to improve video comprehension and instruction following with two datasets: valley-702k and valley-instruct-73k. The experimental results demonstrate a large improvement with the newly curated data.

**Audience:**

Yes

**Claims And Evidence:**

No

**Requested Changes:**

I'll suggest the authors provide more details about the implementation. For example, the use of word "ChatGPT" is confusing and many other details are missing.

**Strengths And Weaknesses:**

Strengths:

1. The paper is easy to follow and well-organized.

2. The proposed method is simple but effective.

3. The curated dataset made a contribution to the community, which could be leveraged to boost the multi-modal model training with high-quality data.


Weaknesses:

1. The biggest issue to me is this paper lacks novelty as the core idea essentially replicate swork from image-based multi-modal research (e.g., LLaVA, InstructionBlip). The only contribution seems to be the datasets but similar datasets have already been introduced, such as VideoChat and Video-ChatGPT.

2. The authors mention that the data was annotated using ChatGPT. Could you clarify which version of GPT was used? Was it GPT-3.5, GPT-4, or GPT-4o?

3. Additionally, the authors state: “When using chain-of-thought (by adding ‘you need to think step by step’ to the system prompt), the performance of Valley is generally comparable to or even better than GPT-3.5.”
To my knowledge, GPT-3.5 only has text processing capabilities. Could you provide more details on how this baseline was evaluated? Further clarification is needed.

---

> ### Author Response · Authors · 2025-01-15
>
> We thank Reviewer 5Umb for the comments that help improve this paper. Next, we provide a point-to-point response to each comment.
>
> **W1.** Regarding the novelty of our proposed Valley
>
> **Response:** We propose employing a simple linear projection layer as a bridge between the video and text modalities, while similar works mainly adopt the complex Q-former structure In early 2023. Besides, we specifically design three temporal modeling modules to improve video understanding.
>
> **W2 & Q1.** Confusion for the use of ChatGPT
>
> **Response:** We have double-checked and revised the manuscript to clearly indicate the version of GPT.
>
> **W3 & Q1.** Clarification of evaluation details.
>
> **Response:** This statement is on the results for the Science-QA dataset, which is a textual understanding dataset. We have provided more evaluation details in the revised manuscript.

---

### Decision · Action_Editor_zEsD · 2025-02-01

**Recommendation:** Reject

**Comment:**

The paper presented a method for joint video and language training. The reviewers think the paper is well-written, and the method is simple but effective. However, there are several major concerns from the reviewers: (1) the collected dataset requires high quality and the details of the collection have not been released; (2) the experimental results and analysis are not solid; (3) some of the claims made by the paper are not accurate. Considering these issues, AE decided to reject the paper. If possible, the authors are welcome to polish the paper and submit it again to the journal.

**Audience:**

Not really. Most of the finds of this paper are not new.

**Claims And Evidence:**

Some of the claims made by the paper are not correct (mentioned by R1 and R3). Some of the claims are not fully supported by the facts as many of the details are missing.

**Resubmission Of Major Revision:**

The authors may consider submitting a major revision at a later time.